# Reliable retrieval is intrinsically rewarding: Recency, item difficulty, study session memory, and subjective confidence predict satisfaction in word-pair recall

**Linus Holm\*, Michael Wells**

Department of Psychology, Umeå University, Umeå, Sweden

\* linus.holm@umu.se

**Data Availability Statement:** The data used in the study can be found in this open science depository: https://osf.io/dsvp9/?view_only= ccba9af99d8543abaa82fd4a32952fbd.

## Abstract

The recall of a distant memory may appear satisfying and suggest successful retrieval is inherently rewarding. If the brain incentivizes retrieval attempts on the prospect of an internal retrieval reward, then the desire for that reward might natively reinforce declarative memory access. But what determines the level of retrieval satisfaction? We tested the idea that retrieval attempt uncertainty drives retrieval satisfaction. For instance, the more distant the memory, the more satisfying should it be to successfully retrieve it. Alternatively, the brain issues rewards based on the level of confidence in recall independent of the recall achievement. If so, then more confident retrieval is also more satisfying. In an online experiment containing five Swahili-English word pair study sessions spaced across one week, we tested 30 English-speaking participants' recall satisfaction and memory confidence during learning as well as in a final cued recall test. We hypothesized that retrieval satisfaction should either increase or decrease with retrieval uncertainty as indicated by time since encoding, and how little in overall they recalled from the session. We found that retrieval satisfaction decreased with time since encoding and with study session retrieval performance. Moreover, we found that retrieval confidence and satisfaction ratings were highly related in the experiment. We also found a reliable interaction between confidence and word difficulty indicating that confidently recalled difficult items induced more satisfaction. Thus, the brain appears to reward both retrieval confidence and to a lesser extent, fruitful retrieval effort. Our findings may explain seemingly irrational self-regulated study behavior such as avoiding learning-efficient but difficult training protocols, as effects of a system rationally seeking to accrue intrinsic cognitive reward.

## Introduction

On repatriation after several years abroad, I failed to retrieve my old lab computer password. I knew the password contained an unintuitive series of characters, numbers and signs. One evening a few days later however, I had a moment of clear recall, and happily accessed the computer the day after. Not only did I experience relief, but also accomplishment suggesting

**Funding:** LH 2019-02997 Swedish Research Council https://www.vr.se/ The sponsor played no role in study design, data collection, analysis, decision to publish or preparation of the manuscript.

**Competing interests:** we the authors declare that no competing interests exist.

reward circuitry was at play. Is it possibly the brain rewards itself for successful recall? Several fMRI studies support the idea that recall may be associated with increased activity in striatal regions sensitive to reward, even when recall is not extrinsically motivated and independent of the memory content [1–4]. Specifically, a retrieval attempt constitutes a goal directed action, and the outcome of this action may drive intrinsic reward signals, an idea captured by the notion of goal attainment in recall [3]. The implications of the brain rewarding itself for reaching cognitive goals seem vast, and many questions arise regarding the brain's policy for distributing such intrinsic rewards. The purpose of this study was to test what drives the intrinsic reward quantity in successful *recall.*

As a point of departure, we consider why the brain might reward itself for successfully retrieving memories. One possibility is that the retrieval reward policy acts to improve memory by guiding retrieval attempts to memories with higher posterior accessibility once retrieved. There is ample empirical evidence that attempting and succeeding at retrieving memories increases future chances of recall [5, 6]. Presumably the memories that stand the most to gain from such retrieval practice are the faint or uncertain memories [7, 8]. If so, prior uncertainty of the to-be recalled memory should drive the intrinsic reward issued by successful recall. Several factors influence confidence and retrieval performance including time since encoding [9, 10], attention at encoding (e.g., [11]), application of learning strategies during encoding (e.g., [12, 13] and sleep (e.g., [14, 15], to cursorily mention a few. Moreover, several experimental studies on curiosity [16–19] suggest that uncertainty reduction drives a learning reward anticipated from information seeking. Might meta-memory act in a similar way albeit directing attention to internal information search? If so, it then follows that the more uncertain the memory is at outset of an attempt, the more rewarding should it be to successfully retrieve it. This would predict that the further back in time, and the less focused the encoding, the more satisfying should it be to recall the material from the event. Under this account, retrieval reward is determined by a positive prediction error of the retrieval attempt.

Another possibility is that the brain rewards retrieval *proportional* to the confidence in the retrieved memory. Recent events are presumably more relevant to the organism [20, 21]. Retrieving recent memories involve more detail [22] and those memories are typically retrieved with higher confidence, as decay or forgetting has not yet diminished its content or access. Confidence might then act as a relevance cue guiding retrieval attempts to recent or highly accessible events. In addition, the cognitive effort of attempting to retrieve a recently encoded item is typically smaller (e.g., [23], which motivates higher satisfaction as a reflection of higher retrieval attempt utility. The view that confident retrieval activates striatal reward circuitry has received some support from brain imaging studies, suggesting that confident recollections activate striatal reward circuitry more readily than low confident recollection [24]. Of particular note, Clos and colleagues [1] found that the satisfaction experienced in image recognition and memory confidence ratings were strongly associated. Moreover, almost all confidence-related striatal activity in their study was explained by recognition satisfaction as tested via pleasantness ratings.

Here we expand on Clos and colleagues' [1] findings by parametrically varying retrieval certainty via memory retention, whereas Clos and colleagues employed a single immediate study session followed by a recognition test. Moreover, we employ word-pair retrieval rather than recollection responses in a recognition test. Furthermore, in addition to confidence ratings, we use study session memory and item-level estimates of difficulty as predictors of retrieval satisfaction. For instance, retrieving what the teacher said last week may be challenging because attention was shared with a smartphone during the session, or because specific facts taught were difficult to comprehend.

Rewards as conventionally defined typically target behavior rather than subjective experience [25]. Therefore, the experience of pleasure or satisfaction does not necessarily follow

from the reception of a reward, which has led to the distinction between *liking* and *wanting* [26]. In the *absence* of satisfaction, the system may still have been rewarded, but there is always some reward at play in the *presence* of satisfaction. Thus, the presence of satisfaction following a reward constitutes a sufficient criterion for the presence of reward [26]. In the present study we use satisfaction ratings as a measure of experienced reward.

For metamemory to be adaptive, memory confidence judgments need to be tied to objective performance. Similarly, for the brain to be rational in its intrinsic reward-policy, the signal it uses to reward cognition must be tied to both objective measures of reliability as well as (and to a larger extent) to the brain's assessment of that reliability. To induce variability in retrieval performance and confidence, we employed a spaced Swahili-English word-pair learning task, which involved five study sessions per participant. The study sessions were administered 7, 3, 2, 1 day(s), and immediately preceding the final retrieval test, respectively. To verify learning of all items during the study sessions, each study session involved an immediate retrieval phase with re-study and re-retrieval until each paired associate word was correctly recalled exactly once. We asked participants to rate recall confidence, judgments of learning and satisfaction during the recall phase of each study session and again in the final retrieval session on the 7th day.

In addition to decay as a source of memory retrieval uncertainty, there may be uncertainty related to intrinsic difficulties in encoding a word-pair. Therefore, we predicted that word-pair difficulty should predict retrieval satisfaction just like encoding session spacing with respect to recall. The immediate recall and re-study structure of the study sessions allowed us to estimate word-pair difficulty using Rasch [27] item response methods. The method assumes that two latent factors determine the probability that a person responds correctly to a test-item; individual skill and item difficulty, respectively. The two latent factors are estimated from a set of test scores by a set of test takers. We predicted that word-pair retrieval difficulty would induce uncertainty in the same way as retention time and therefore allowed us to test the same hypotheses based on performance and ratings during the study sessions as employed in the final recall session.

Finally, many recent curiosity studies posit that curiosity is driven by an anticipated learning reward [16–19]. One might view a recall attempt as a search for an uncertain memory and therefore predict that more uncertain or distant memories would also induce greater curiosity–as well as satisfaction upon recall. We therefore employed the complementary methods of assessing curiosity via curiosity ratings during study sessions and via willingness to wait for feedback in the final recall test. The latter method utilizes aversion to waiting to quantify the subjective value of information [16–19] here defined as the likelihood of waiting for feedback. We therefore imposed a waiting time conditioned on response feedback request in the final recall test. Based on current curiosity theory, we expected to see higher curiosity ratings and likelihood of waiting following uncertain word-pair responses. Moreover, Metcalfe and colleagues [28] found that so called "tip-of-the-tongue" phenomenon were more common preceding requests for answers to trivia questions. We therefore also asked participants to state whether they experienced a tip-of-the-tongue state in the final recall test and hypothesized that they would be more likely to request to see the English translation following a tip-of-the-tongue state.

## Methods

### Participants

We recruited participants through the online platform Proflific.co. Potential participants were screened according to their country of birth and residence so that only countries where

English is spoken as primary language were included; this restricted recruitment to the United Kingdom (UK), Ireland, The United States of America (USA), Canada, New Zealand and Australia. Additionally, prospective participants were screened according to their self-reported primary language as to only include those persons who reported English as their first language. Finally, users were only able to take part if their computer hardware met the minimum requirements to access the experiment. Participants were offered £5 per session and a bonus £10 for completion of the study to incentivize continued engagement with the project. Out of 70 participants who completed the first session, a total of 30 participants completed all sessions of the experiment and met our inclusion criteria. Of those who completed the study, 8 identified as male, and 22 identified as female. The participants' average age was $M = 29.9$ years, ($SD = 9.4$). This project was approved by the Swedish ethics authority (2020–02262) and conducted in compliance of the declaration of Helsinki. Participants were recruited June 2021. The authors did not have access to any data pertaining to the identity of the participants.

## Instruments and materials

**Word lists.** A master word list containing 100 Swahili words and their English translations was produced based on the following criteria. Words in both Swahili and English were restricted to having between 4 to 6 characters in length, were proper nouns, and where in English the words had no obvious additional functions or meanings (i.e., restricted polysemy). Moreover, no Swahili words that had their origins in English were used. The English translations were further subjected to screening through the use of the British National Corpus (BNC) [29]. Only those words that had a usage greater than 5 words per million were selected. Once compiled, the master word list was split into 5 groups of 20 words according to each word's usage per million in English to keep lists as similar as possible.

**Experimental resources.** The experimental resources used in this study were produced using the PsychoPy2 software package [30] and were hosted online at https://pavlovia.org/, that interlink with the Prolific platform.

## Procedure

Participants were recruited in five separate groups over a period of 48 hours. Each group of participants was assigned to one word-pair list order (latin-squared list orders across groups) This staggered recruitment approach was necessary due to a limitation of the Prolific platform whereby it was not possible to stop the same eligible participant from signing up to multiple cohorts within the same experiment if all cohorts were started simultaneously.

Upon study admittance, the participant was directed to Pavlovia.org where the experiment was being hosted. The experimental environment when loaded initially began with a welcoming screen containing instructions outlining expectations, remuneration and ethical considerations including the General Data Protection Legislations (GDPR), anonymization of data and the right to withdrawal at any point without consequence. Consent was obtained by clicking a consent button that also initiated the experiment. Hitting the escape button and hence exiting the experiment environment at any point during the trial was interpreted as withdrawal of consent.

After consent was obtained, a blanking screen containing a crosshair was presented (0.3s), followed by an instruction screen. The encoding loop then began. The encoding screen consisted of a crosshair, with a word pair drawn randomly from the corresponding word list for that session; the Swahili word was presented on the left and the English word on the right. Participants had 20s to memorize the word pair, after which a new pair was randomly presented

from the remaining 19 words in the list until all 20 words had been shown for the session. An instructional screen was then presented, followed by a blank screen (0.3s) after which the test phase of the encoding loop began.

In the study session test phase, participants were presented with the Swahili word with an empty text box next to the word, and participants were instructed to enter the English translation in the box within a response deadline of 20 s. The time limit was used to discourage individuals from checking translations online. Upon the 20s elapsing, users were led to a screen that asked for feedback in the form of a questionnaire with slide bars allowing for a continuous response. Participants were asked "How confident are you that you remembered correctly?" (-1 = Unconfident, 1 = Confident), "Are you curious to see the correct English translation?" (-1 = Uncurious, 1 = Curious). Moreover, participants were asked to make a Judgment of learning by the question: "How confident are you that you can remember this information one week in the future?" (-1 = Unconfident, 1 = Confident) and, finally we assessed reward via ratings to the question "How satisfied do you feel with your answer?" (-1 = Not Satisfied, 1 = Satisfied). Upon answering all 4 questions, participants were allowed to click next, moving them onto the next random word pair with the recall loop.

Once all 20 pairs had been responded to, the environment checked responses for similarities according to the Levenshtein's distance algorithm [31]. The Levenshtein distance is a string metric that expresses the difference between two strings in terms of how many basic operations such as insertion, deletion and substitutions are needed to transform one string into the other. An edit distance of $\leq 2$ was used to determine whether a word was answered correctly or not. Incorrect answers were recorded into a personalized list for each participant, and they were prompted to repeat the entire session with just these words until all words had been correctly recalled once. Upon successful completion, an ending screen was presented to remind them that they would be called back later. A button was used to end the experiment at this stage to record that these additional instructions had been read.

Each participant was then called back at day 4, 5, 6, and 7 after recruitment to perform the same procedure but with different lists to those previously studied. In the final (5th) encoding session on the 7th day, participants were subjected to the same procedure with their final word list, however an additional final recall test was added instead of an exit screen.

At the final recall test, after a brief set of instructions, all 100 words learned during previous encoding sessions were presented in turn. As in the testing phase of the encoding loop, the Swahili word was presented on the left, a crosshair in the middle and a user editable textbox on the right for the English translation. After each word recall attempt, the following questions were asked. "How satisfied do you feel with your answer?" (-1 = Not Satisfied, 1 = Satisfied), "How confident are you that you remembered correctly?" (-1 = Unconfident, 1 = Confident) and, "How Certain are you that you have seen this word before?" (-1 = Uncertain, 1 = Certain). Users answered using a sliding scale. Moreover, a prompt asked "Did you experience a "tip of the tongue" effect?" which was answered with yes/no radio buttons. Finally, the question "Would you like to see the answer?" was answered with a yes and no button. Upon pressing yes, the user was presented with text explaining a time gap before the answer was presented. After a 5 s waiting time elapsed, the Swahili word that they wished to know the English translation to was presented on the left, and the translation on the right of a crosshair in the center of the screen. By clicking next, the loop continued presenting the next word pair in the 100-word list. If a user clicked no, the loop was continued without the addition of the 5s wait and answer screen. Once all 100 words had been responded to, the user was presented with a sequence of questionnaires regarding their experience of the experiment. The results of these surveys are outside of the scope of the present study.

## Results

### Attrition analysis

Across the five sessions, 70, 40, 34, 31 and 30 participants completed the corresponding session. Due to this substantial attrition, we tested whether data from the those who dropped out differed in any substantial way from those participants who completed the study. Of the participants completing the study, 22 identified as female and 8 as male. Of those who did not complete all sessions, 20 identified as female and 11 as male (note that some participants who dropped out did not provide a complete record hence fewer gender responses than number of participants who did not complete the study). We tested the gender distribution difference with Fischer's exact test which yielded $p = .58$ and an odds ratio of 1.5. Thus, the numerically higher tendency for females than males to complete the study was not reliable. The average age of participants who did not complete the study was $M = 35.1$ years ($SD = 11.5$) and was numerically higher than the average of those participants who completed the study ($M = 29.9$, $SD = 9.4$). The difference was not statistically reliable as tested with an unequal samples t-test, $t(66) = 1.98$, $p > .054$. We also tested whether the participants who dropped out differed substantially in terms of task performance and ratings. The proportion correct recalls in the first attempt of the first study session was rather similar between participants who did not complete ($M = .44$, $SD = .20$) and those who completed the study ($M = .47$ $SD =, 21$), $t(65) = .38$, $p = .78$. We also compared the association between z-scored satisfaction ratings and confidence from the first study session for participants who dropped out of and completed the study, respectively. Please note that this only contains 20 trials per participant which we further restricted by limiting the analysis to correct first recalls. We separately tested the models *Satisfaction ~ Confidence + 1| participant* against the null model without the fixed term, for participants who completed and did not complete the study, respectively. With 282 observations of participants who completed the study, we found a reliable association between confidence ratings and satisfaction, $b_{confidence} = .59$, $p < .0001$, which was better supported than the null model, *Likelihood Ratio (LR)* $= 39.3$, $p < .0001$. Thus, satisfaction increased with confidence for correct recalls among participants who completed the study, echoing the results presented above albeit here with a more limited data set. Importantly, we found a similar effect also for participants who did not complete the study $b_{confidence} = .73$, $p < .0001$, which was better supported than the null model, $LR = 49.0$, $p < .0001$.

### Encoding session results

Participants recalled 61% of the items in their first recall attempt and never needed more than five repetitions until they correctly recalled an item. Performance during study sessions varied in terms of number of repetitions until all items were recalled correctly. Average number of repetitions across the study sessions 1:5 were $M = 1.74$ (.090), 1.34 (.13), 1.37 (.16), 1.56 (.21) and 1.37 (.13), respectively (*SD* in parentheses). The difference in number of repetitions until criterion was tested in a repeated measures ANOVA indicating a statistically reliable difference with $F(4,112) = 14.9$, $MSE = .016$, $p < .0001$.

The Satisfaction ratings of the first recall attempts of each word were higher following correct ($M = .67$, $SD = .18$) than incorrect retrieval, ($M = 0.19$, $sd = .19$), pairwise $t(29) = 12.8$, $p < .0001$. Similarly, first confidence ratings following correct retrieval attempts ($M = .75$, $SD = .11$), were higher than following incorrect retrieval attempts ($M = .15$, $SD = .12$), pairwise $t(29) = 23.8$, $p < .0001$. Excluding omissions (i.e., trials where no retrieval response was entered) from the comparison yielded qualitatively similar results. Specifically, satisfaction ratings for incorrect responses were $M =, 39$ ($SD = .23$), and a pairwise comparison yielded $t(29) = 7.9$, $p < .0001$. Similarly, confidence ratings to incorrect retrieval responses were $M = .35$ ($SD = .21$)

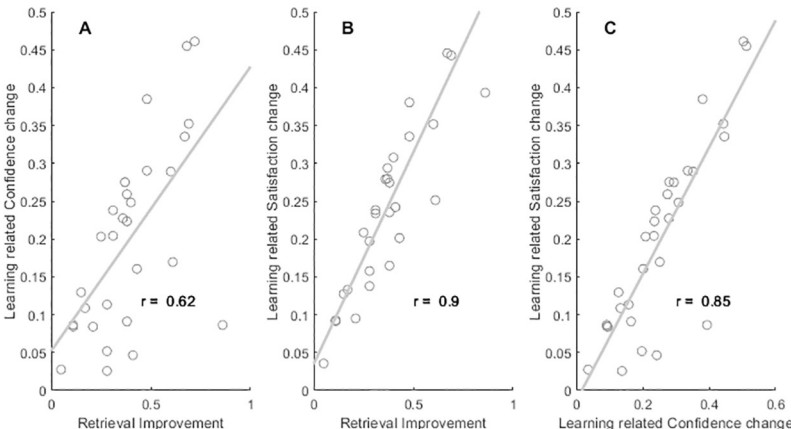

**Fig 1. Study session learning and ratings.** Panel A shows the association between within-participant average confidence rating and performance change from first to last retrieval attempt (i.e., 1—proportion correct responses in first attempt). Panel B shows the association between within-participant average satisfaction rating and performance change from first to last retrieval attempt. Panel C shows the association between within-participant average satisfaction and confidence rating *change* across first and last retrieval attempt during encoding. Pearson product moment correlation coefficients entered in each panel, respectively, all at $p < .01$.

and yielded a statistically reliable difference, $t(29) = 9.3$, $p < .0001$. This then suggests that learning (i.e, from failed to successful retrieval attempt across repetitions) should result in increased retrieval confidence and satisfaction.

As seen in Fig 1, participants' retrieval confidence and satisfaction increased between first and final retrieval attempt. The change in confidence and satisfaction across first and final retrieval attempt by design involved a change in performance as items were re-presented until successfully recalled. Importantly and immediately visible from Fig 1, no single participant displayed *decreased* average confidence or satisfaction following performance improvement, which would have been displayed as negative values along the respective y-axes.

To test whether pre-retrieval attempt uncertainty predicts retrieval satisfaction also when restricting the analysis to correct recalls, we conducted a Rasch analysis [27] based on first retrieval attempt data and used the estimated item-by-participant performance as a proxy for pre-retrieval uncertainty to predict study-session retrieval-satisfaction. The association between Rasch estimates of participant average performances and empirical participant averages were $r^2 = .90$ and the association between Rasch estimates of average item performance and empirical item average performance was $r^2 = .97$. If memory reliability drives retrieval satisfaction, we should see higher satisfaction ratings with higher confidence ratings for correctly recalled items. If instead uncertainty drives satisfaction for successfully recalled items, we should see higher satisfaction ratings with lower expected performance and confidence. As seen in the Fig 2A, Rasch estimates of performance were generally positively associated with satisfaction ratings. This pattern of results was even more pronounced in the association between confidence ratings and satisfaction (Fig 2C). We subjected these results to mixed effects linear regression analysis with estimated performance as a fixed effect and participant as a random variable. All rating data from correct recalls were centered (z-scored) before analysis as were all other ratings tested in mixed effects models in the study. The model Satisfaction ~ Rasch performance + Rasch performance | participant yielded $b_{Rasch} = 0.35$, $t = 2.73$, $p < .01$ with 1820 observations. The model with Rasch performance as a fixed effects term was reliably better supported than a null model without the fixed effects term, $LR_{fixed} = 8.0$, $p < .01$. We tested the corresponding effect replacing satisfaction with

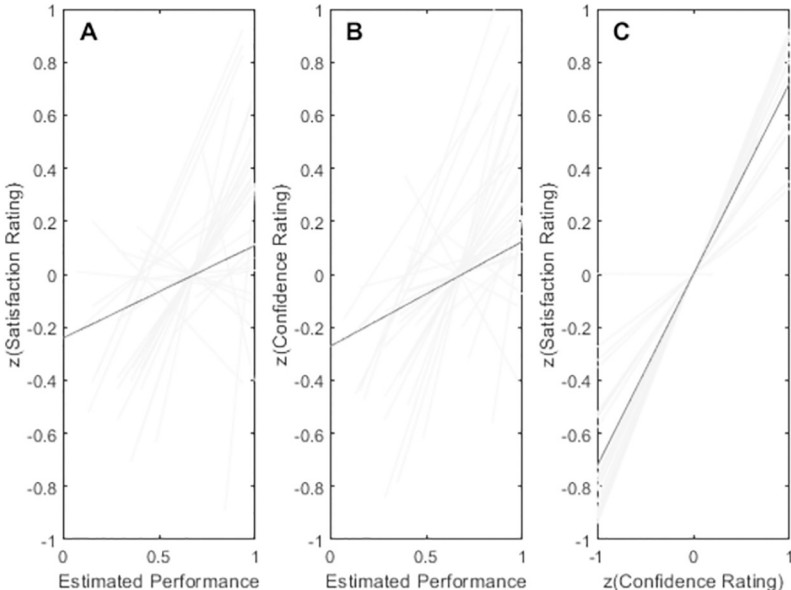

**Fig 2. Performance and rating associations.** Panel A shows the satisfaction ratings as a function of Rasch estimated performance. Panel B shows confidence ratings as a function of Rasch estimated performance. Panel C shows association between satisfaction and confidence ratings. Light grey lines reflect linear fits to individuals. Dark grey line indicates fixed effects linear fit. Data restricted to z-scored ratings to correctly recalled items in first recall attempt at study.

confidence and found that $b_{Rasch}$ = .40, $t$ = 3.2, $p < .01$. The model with Rasch performance as fixed effects term was reliably better supported than a null model without the term, $LR$ = 10.0, $p < .001$. We also tested response confidence as a predictor of satisfaction by running confidence ratings as a fixed effects term in a linear mixed effects model and found a reliable association between satisfaction and confidence ratings with $b1$ = 0.72, $t$ = 17.0, $p < .001$ indicating that more confidently recalled items are also experienced as more satisfying. However, the possibility still exist that ease of recall and confidence may interact with satisfaction such that items that appeared difficult to recall (as indicated by the Rasch performance estimate) yet were confidently recalled, appeared as *more* satisfying than an equally confident recall of an item that was easier to retrieve. We therefore tested the model Satisfaction ~ Confidence + Rasch performance + Confidence x Rasch performance + 1 | participant. Indeed, the interaction was reliable negative at $b_{Confidence \ x \ Rasch}$ = -.25, $t$ = -2.83, $p < .01$, the likelihood ratio for the interaction model over a model only containing the fixed effects (i.e., Satisfaction ~ Confidence + Rasch performance + 1 | participant) was $LR$ = 8.0, $p < .01$ suggesting that for equal confidence, more difficult items (i.e., low Rasch performance) induced higher satisfaction ratings when successfully recalled.

**Curiosity at study.** We used item difficulty as estimated via Rasch methods and subjective confidence as predictors of curiosity ratings during study. So as not to confound interpretations we restricted the analysis to responses to the first retrieval attempt at study and included both correct and incorrect recall responses. For the model Curiosity ~ Rasch performance + Rasch performance | participant. With 3000 observations, we found no support for a reliable fixed effect of Rasch performance on curiosity ratings, $b_{Rasch}$ = -.19, $t$ = -1.85, $p$ = .064. Also, when replacing Rasch performance with confidence ratings in the fixed effects model, we found no reliable effect of confidence on curiosity ratings, $b_{confidence}$ = -.17, $t$ = -1.91, $p$ = .056.

## Final recall test results

Satisfaction ratings were statistically reliably higher for correct ($M$ = .72 $SD$ = .19) than incorrect ($M$ = .18, $SD$ = .20) recall responses, $t(29)$ = 12.6, $p$ < 0001. The pattern of results remained also when restricting the analyses of incorrect responses to responses with letter entries (i.e. removing omissions), $t(29)$ = 7.90, $p$ < .0001. Similarly, confidence rating were generally higher following correct ($M$ = .71, $SD$ = .20), than incorrect responses ($M$ = .15, $SD$ = .14), $t(29)$ = 15.5, $p$ < .0001. Restricting the comparison to incorrect responses with letter entries only also yielded a reliable difference, $t(29)$ = 9.28, $p$ < .0001.

As seen in Fig 3A, average memory performance exhibited a decay pattern in time since encoding. Restricting the analyses to correctly recalled items only, the figure suggests that both satisfaction and response confidence decreased as a function of time since encoding. Thus, the closer in time, the more likely was the paired associate to be recalled, it was recalled with higher certainty, and the successful recall response was rated as more satisfying.

We tested whether time since encoding predicted recall performance in a mixed effects regression with log time since encoding as a fixed effect. With 3000 observations, we found that $b_{log(time)}$ = -0.062, $t$ = -6.43, $p$ < .001 indicating that recall performance rates reliably decreased with time since encoding. Furthermore, we employed a mixed effects linear regression test of retrieval satisfaction for correctly recalled items as a function of log time since encoding as fixed effect and with participant as random variable. With 1172 observations, the fixed effect of log time since encoding was $b_{log(time)}$ = -0.027, $t$ = -2.19, $p$ < .05, with $LR$ = 9.19, $p$ < .001. The corresponding analysis with time since encoding as a predictor of confidence rating yielded $b_{log(time)}$ = -0.03, $t$ = -2.73, $p$ < .01 with $LR$ = 6.30, $p$ < .05. Finally, noticing that the proportion recalled items varied substantially across sessions and between participants (see Fig 3A), we tested session average recall score (henceforth referred to as session memory) as a predictor of correct recall satisfaction and response confidence, respectively. For satisfaction, we found that session memory $b_{session\ mem.}$ = 0.50, $t$ = 3.35, $p$ < .001 and $LR$ = 9.19, $p$ < .01. For confidence ratings, session memory $b_{session\ mem.}$ = 0.59, $t$ = 3.88, $p$ < .0001 and $LR$ = 12.0, $p$ < .0001. While these results suggest that objective measures of memory reliability determine

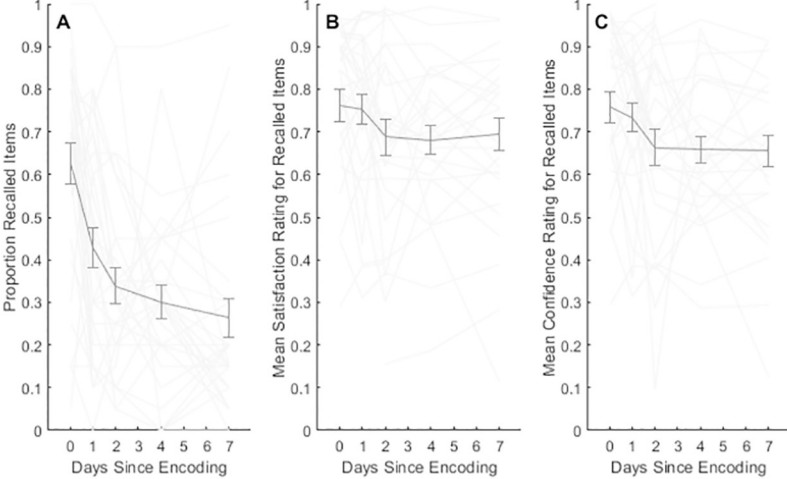

**Fig 3. Forgetting and recall ratings.** A) Average final recall test performance as a function of time since encoding B) Average final recall test satisfaction ratings for correctly retrieved items as a function of time since encoding C) Average final recall test confidence ratings for correctly retrieved items as a function of time since encoding. Error bars reflect 1 SEM. Faint grey lines show individual participant performance across the five sessions.

recall satisfaction, there is also a possibility that ease of recall and confidence interact to determine retrieval satisfaction such that e.g., recalling something perceived to be difficult to encode confidently is more satisfying than confidently recalling items that appeared easy to encode. We therefore tested Satisfaction ~ Confidence + Rasch performance + Confidence x Rasch performance + 1 | participant, and found that the interaction indeed was negative with $b_{Confidence\ x\ Rasch}$ = -.22 and statistically reliable at $t$ = 1.99, $p < .05$. The model was also reliably better supported than the null model without the interaction term, $LR$ = 4.0, $p < .05$. In addition to item difficulty, retention time might also interact with confidence following the same reasoning. For instance, distant confidently recalled memories may yield a stronger sense of satisfaction than the recall of more recent confident memories. We therefore also tested the model Satisfaction ~ Confidence + log(Time) + Confidence x log(Time) + 1 | participant. However, we found no support for a reliable interaction; $b_{Confidence\ x\ log(time)}$ = 0.01, $t$ = 1.57, $p$ = .12.

To test meta-memory calibration, we also submitted participants' final judgments of learning (JOL) from the study sessions as a predictor of recall in a mixed effect logistic regression with individually z-scored JOL rating units at study as a fixed effect. With 3000 observations, we found that judgment of learning ratings predicted recall in that $b_{JOL}$ = 0.21, $t$ = 2.44, $p < .05$ and the model was reliably more likely than the null model without the fixed effect, $LR$ = 4.66, $p < .05$. Thus, participants were about 23% more likely to recall a word-pair at the final test per z-scored JOL rating at study.

**Curiosity at final recall test.** To investigate curiosity at the final recall test, participants could request to see the correct translation at a waiting cost. Likelihood of requesting feedback thus served as a measure of curiosity. We tested whether time since encoding predicted the likelihood of requesting feedback in a mixed effects logistic regression that yielded $b_{log(time)}$ = .08, $t$ = 2.38, $p < .05$. However, this analysis would confound number of incorrect or failed recalls with time since encoding. A stricter analysis would be limited to incorrect recalls only. The restricted analysis without correct recalls yielded $b_{log(time)}$ = .029, $t$ = .7, $p$ = .48. For incorrectly recalled words, likelihood of requesting feedback as a function of time since encoding was not reliable. Furthermore, we replaced time since encoding with confidence ratings and found $b_{confidence}$ = 1.46, $t$ = 1.14, $p$ = .25. Thus, confidence was not associated reliably with likelihood of requesting feedback.

We also tested whether the likelihood of requesting feedback was higher when in a so-called tip-of-the-tongue state (TOT, Metcalfe et al. 2017). Overall, participants reported being in a TOT state in 34% of the trials. Participants were numerically slightly more likely to request feedback when in a TOT state ($M$ = .61) compared to not in a TOT state ($M$ = .52) but the difference was not statistically reliable, $t(29)$ = 1.84, $p$ = .076.

## Discussion

We found that factors known to influence recall uncertainty including item difficulty, retention interval, and study session average memory performance directly affected recall satisfaction, and consistently demonstrate that recall satisfaction increases with recall confidence. Thus, it appears as if the brain promotes reliable retrieval by rewarding itself based on the subjective memory confidence signal. We found a small interaction effect between confidence and estimated performance, suggesting that for difficult items, high confidence bestows a bonus on recall satisfaction. But taken together, the results strongly point to recall satisfaction being largely determined by subjective response reliability.

Our findings offer a potential explanation for Bjork's [7] concept of "desirable difficulties" according to which less efficient learning protocols are often perceived as more efficient and appealing. For instance, spacing can produce more efficient category learning than massing

yet be perceived as less efficient [32]. Moreover, Baddeley and Longman [33] found that spreading typing lessons across a longer period instead of massing training produced faster learning per hour of practice but was less preferred over massing. Similarly, re-studying material is typically favored over taking a test on the studied material whereas testing usually leads to superior long-term retention [5]. Learners' preferences in these circumstances seem rational under the assumption that they evaluate the reliability of the immediate learning outcome and aim to amass cognitive satisfaction by increasing their memory confidence.

We exposed participants to multiple ratings both in the study and final recall sessions, which might have introduced a risk that participants habitually responded to the different sliders, instead of indicating their actual judgments. In that case we would expect to see similar associations between curiosity and confidence ratings as we obtained from the confidence and satisfaction ratings, as participants would then respond the same to all items. However, the association between study-session curiosity and recall confidence was barely reliable, in stark contrast to the association between recall satisfaction ratings and recall confidence. We therefore interpret our results as valid reflections of participants' judgments to the questions asked rather than the spurious result of habitual responding. The critical reader is invited to test all other associations in our published data to dispel other possible remaining concern pertaining to the validity of our methods and core finding on the association between recall satisfaction and confidence.

The attrition rate was rather high in the experiment as only 30 out of 70 recruited participants completed all five sessions. The group of participants who did not complete the experiment were slightly more likely to be male and a few years older than the participants who completed the experiment. Importantly though, the ones who dropped out did not perform worse in their first study session and displayed the same association between satisfaction ratings and confidence as the participants who completed the study. Nonetheless, the sample that completed the study is strongly gender biased towards female participants. However, we did not set out to test a population-representative sample in this study, and the question of background variables' potential impact on intrinsic motivation for cognition might be better suited for future studies designed for testing representative samples.

Assuming the cognitive satisfaction and confidence association holds, what then might be the adaptive value of a system rewarding itself proportional to the confidence and ultimately the veridicality of recalled memories? Recalled memories may be assumed to guide action directly or indirectly via further thinking, through communication etc. Memories thus constitute information sources for guiding action and the reliability of the information source directly translates to the reliability of the memory trace. In other words, regardless of the validity of the memory, the perceived confidence in the memory should upper bound the information value of the memory. In terms of retrieval focus, recall of recent, well-repeated memories, encoded under full attention, would all have satisfaction utility. Future studies might investigate preferences in non-instrumental recall to test this idea. Is for instance recall in mind-wandering characterized by the retrieval of high confidence memories? Perhaps nostalgia, and frequently revisited past events in casual conversation by old friends also appeal because of the confidence in retrieval associated with the occasions?

We found a small interaction between retrieval confidence and item difficulty with respect to satisfaction. It is potentially a promising finding from a pedagogical view that challenging retrieval attempts may offer a bonus sense of satisfaction. But taken together, our findings point to a native difficulty in a narrow hedonic approach to self-guided study. In the short term, accruing study satisfaction is easier if study time is devoted to study content that is essentially already well encoded (and likely to be recalled with high confidence). On a more optimistic note, restudying content that was poorly encoded substantially increases longer term

satisfaction utility (see e.g., Fig 1) as overall content confidence increases across content repetitions (i.e., re-exposure to retrieval attempts) as does retrieval satisfaction. The poorly rewarded learning sessions pay off in the final exam.

Cognitive uncertainty is not restricted to metamemory but appears in all cognitive domains including problem-solving, decision-making, and perception, as well as memory. The question arises whether cognitive confidence is generally gratifying–and by extension–that reliable cognitive operations are sought for their own sake. If confidence acts as a general motivation signal, it may suggest humans have a general drive for accurate cognition. Similar ideas have been suggested before in the context of working memory [34], but the empirical evidence for the signal that predicts such motivation is currently not well understood. Speculatively, the tendency to off-load working memory using e.g., notes might also be incentivized by the added sense of confidence (and typically, the reliability) of operation that the embodiment entails.

Our results do not support the view that uncertainty is sufficient to drive curiosity. Neither the curiosity ratings in the study sessions, nor the propensity to request feedback at final test were reliably predicted by confidence. For the current learning reward account of curiosity to be true [35–37] would require that the brain makes a more sophisticated forecast on its ability to encode new information than merely an evaluation of its current information uncertainty. Arguably, tip-of-the-tongue states should indicate potential to update contingent on feedback [38], yet we found no reliable support for that notion. Future studies may investigate whether e.g., expected information gain better predicts curiosity. It is also possible that the learning reward argument in curiosity theory is incorrect or at least that the hedonic 'wanting' [26] aspect of reward is irrelevant to curiosity.

## Acknowledgments

We are grateful for Patrick Oden's help with arranging the online testing and other members of the Curious Cognition lab for input during the study.

## Author Contributions

**Conceptualization:** Linus Holm.

**Data curation:** Michael Wells.

**Formal analysis:** Linus Holm.

**Funding acquisition:** Linus Holm.

**Investigation:** Linus Holm.

**Methodology:** Linus Holm.

**Project administration:** Linus Holm, Michael Wells.

**Resources:** Linus Holm.

**Software:** Linus Holm.

**Supervision:** Linus Holm.

**Validation:** Linus Holm.

**Visualization:** Linus Holm.

**Writing – original draft:** Linus Holm.

**Writing – review & editing:** Linus Holm, Michael Wells.

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
