## [Decision Letter · Decision Letter 0]

29 Jun 2023

PONE-D-23-11730Reliable retrieval is intrinsically rewarding:

Recency, item difficulty, session performance, and subjective confidence predict satisfaction in word-pair recallPLOS ONE

Dear Dr. Holm,

Thank you for submitting your manuscript to PLOS ONE. After careful consideration, we feel that it has merit but does not fully meet PLOS ONE’s publication criteria as it currently stands. Therefore, we invite you to submit a revised version of the manuscript that addresses the points raised during the review process.

We look forward to receiving your revised manuscript.

Kind regards,

Stergios Makris

Academic Editor

PLOS ONE

Journal Requirements:

Reviewers' comments:

Reviewer's Responses to Questions

**Comments to the Author**

1. Is the manuscript technically sound, and do the data support the conclusions?

Reviewer #1: Partly

Reviewer #2: Partly

2. Has the statistical analysis been performed appropriately and rigorously? 

Reviewer #1: Yes

Reviewer #2: Yes

3. Have the authors made all data underlying the findings in their manuscript fully available?

Reviewer #1: Yes

Reviewer #2: Yes

4. Is the manuscript presented in an intelligible fashion and written in standard English?

Reviewer #1: Yes

Reviewer #2: No

5. Review Comments to the Author

Reviewer #1: PONE-D-23-11730 - Review

Summary

The authors present a study in which they assess whether recall is rewarding. They set out to investigate two mutually exclusive accounts: either distant, more difficult-to-retrieve memories would produce a stronger sense of satisfaction (internal reward) when recall is successful or, alternatively, recent, easier-to-retrieval memories would produce a stronger sense of satisfaction because they do not require great effort. The authors find evidence for the latter account.

The manuscript is well-written, and the findings are very interesting. There are a few points, however, that require more elaboration, particularly in the Results section, before we can recommend the text for publication. Also, we suggest that the authors share, via OSF, all the analysis files, so that others readers can reproduce the results from the data already shared. Below we elaborate on some major and minor points.

Major points

1. In the title and throughout the manuscript, the use of the term reliable and its variants (e.g., “reliable retrieval”) seemed misleading to me. I understand that reliability, in memory research, refers to the accuracy or the “truthfulness” of the recalled information according to some external objective criterion (e.g., in eyewitness testimony research). Reliability may also have very specific meanings in psychology, such as test-retest reliability. However, in this manuscript, the term reliability was used as synonymous with confidence (e.g., “recall confidence”), which refers to the degree of subjective certainty learners assign to their response (i.e., the subjective confidence in their own memory). The interchangeable use of the terms reliability and confidence may inadvertently confuse the reader. To improve readability and keep consistency with the memory literature in general, I suggest replacing the term reliability and its variants with confidence and its variants (e.g., confident retrieval or memory confidence) when the aim is to refer to learners’ subjective judgments.

2. In the Abstract, the authors make the following statement (brackets added by me): “We found that retrieval satisfaction decreased with time since encoding and [also decreased with] session retrieval performance.” This statement seems to be based on results presented (lines 281–285), showing a Confidence × Rasch Performance interaction. However, in the Results (lines 328–329), the authors claim that “… For satisfaction, we found that session performance bsession perf. = 0.50, t = 3.35, p < .001 and LR = 9.19, p <.01.” The term “session [retrieval] performance” seems to imply two different meanings throughout the manuscript. In Rasch analyses, it seems to refer to the estimated item-by-participant performance in the first retrieval attempt across the encoding sessions (based on items and participants parameters estimated with the Rasch model), a proxy of pre-retrieval uncertainty, as stated by the authors (one estimate per participant). In final-recall test analyses, however, “session performance” seems to refer to the observed final-recall performance (by session; five estimates per participant), used to assess whether the empirical final recall as a function of different retention intervals affect retrieval satisfaction and retrieval confidence. These two senses of “session performance” seem to show different relationships with satisfaction. To improve readability, I strongly recommend that the authors make this distinction explicit and, in the Discussion section, discuss the implications of each of these results for the hypotheses presented.

3. Lines 217–220: “Upon pressing yes, the user was presented with text explaining a time gap before the answer was presented. After a 5 s waiting time elapsed, the Swahili word that they wished to know the English translation to was presented on the left, and the translation on the right of a crosshair in the center of the screen”.

- Why did the authors insert a 5-s delay until the presentation of the correct English word? This introduces a “cost of response”, rendering the response “Yes, I am curious” more time consuming than “No, I am not curious". Given the context of a long, multisession experiment, in which participants are already tired, introducing such 5-s cost is likely to reduce the number of “Yes” responses. Therefore, we suggest that the authors explain why they have chosen to include the 5-s delay and how it could have affected the results and conclusions of their study.

4. Lines 231–232: “Participants recalled 61% of the items in their first recall attempt and never needed more than five repetitions until they correctly recalled an item”. One characteristic of the adopted within-subjects design is that it confounds the time since encoding with the number of intervening lists until the final test (which, speculatively, may have led to a buildup of proactive interference). Did the initial recall reported in the text differ across the five sessions? Please explicitly state this information in the text.

5. Lines 232–235: The authors compared satisfaction and confidence between correct and incorrect retrieval attempts. Among the incorrect attempts, what was the rate of response omissions? If the omission rate was high, I believe it is important to restrict the analyses only to incorrect attempts with responses, as it would not be surprising for satisfaction and confidence to be higher in correct attempts if the incorrect attempts include many omissions.

6. Lines 247–248: “As seen in figure 1, participants’ retrieval confidence and satisfaction increased between first and final retrieval attempt”. Given the dropout procedure adopted in the study, I wonder how the authors defined “final retrieval attempt.” For example, if for a given participant, the last test repetition led to correct recall of the remaining five items, did only those five items contribute to the measures of confidence and satisfaction? Or, alternatively, did the final attempt of each item contribute to the measures of confidence and satisfaction? Could you explain this point?

7. The sample size (n = 30) is quite small for estimating Rasch models. Some points:

a) Please discuss the consequences of the small sample size on the Rasch model estimates (e.g., how would that impact on the relationship between item difficulty and satisfaction)?

b) Rasch model estimates were used as indices of item difficulty. Nelson and Dunlosky (1994) provide norms of recall difficulty for a sample of 100 Swahili-English word pairs. Why did the authors decide to estimate item difficulty from Rasch models rather than from Nelson and Dunlosky’s norms? As a suggestion, the authors could correlate their Rasch model estimates with Nelson and Dunlosky’s (1994) norm scores as a means of validating the items’ difficulty estimates.

c) Additionally, the authors could correlate item difficulty estimates from Figure 2 to the proportion recalled in the final test from Figure 3 (left panel) as an attempt to validate independently their estimates of the Rasch models.

8. Model comparisons (part 1). The excerpts below refer to lines 267–285. In this section of the Results, the authors describe a series of mixed-effect model comparisons to support the patterns described in the figures. The analyses here (part 1) focus on the fixed-effect terms Rasch performance (item difficulty) and Confidence and their interaction. For ease of reference, we will use numbered equations to refer to each mixed-effect linear regression model and the text passage it refers to.

The main question we have here is whether the full model, with the fixed effects of Confidence, Rasch performance and their interaction (Equation 6) is significant when compared to simpler models (Equations 3, 4 and 5). If it is, the role of the interaction should be emphasized in the Abstract and in the Discussion, as the result implies that that Satisfaction is influenced by response Confidence, item difficulty and the Confidence × Rasch performance interaction (for a given level of confidence, correct responses to more difficult items are associated with higher satisfaction).

Below we reproduce the models and the corresponding text to elaborate on our issue regarding model comparison. The simplest model assessed was the following:

1. Satisfaction ~ Rasch performance + (Rasch performance | participant)

(1) “The model Satisfaction ~ Rasch performance + Rasch performance | participant yielded bRasch = 0.35, t = 2.73, p < .01 with 1820 observations. The model with Rasch performance as a fixed effects term was reliably better supported than a null model without the fixed effects term, LRfixed = 8.0, p < .01.” Thus, participants’ satisfaction with their own answers was predicted by item difficulty. The model with the Rasch performance term was compared to the model without that term. The authors also assessed whether confidence could be predicted from Rasch performance. It did:

2. Confidence ~ Rasch performance + (Rasch performance | participant)

(2) “We tested the corresponding effect replacing satisfaction with confidence and found that bRasch = .40, t = 3.2, p < .01. The model with Rasch performance as fixed effects term was reliably better supported than a null model without the term, LR = 10.0, p < .001.” Next, the authors assessed Satisfaction as function of Confidence:

3. Satisfaction ~ Confidence + (Rasch performance | participant)

(3) “We also tested response confidence as a predictor of satisfaction by running confidence ratings as a fixed effects term in a linear mixed effects model and found a reliable association between satisfaction and confidence ratings with b1 = 0.72, t = 17.0, p < .001. The results indicated that the easier the item is to recall, the more satisfying is the experience.” Here, the last sentence refers to item difficulty, when it should refer to Confidence, as Confidence was the fixed-effect term assessed in Equation 3. The authors then proceed to assess whether the interaction term is significant:

4. Satisfaction ~ Confidence × Rasch performance + (Rasch performance | participant)

(4) “However, the possibility still exist that ease recall and confidence may interact in satisfaction such that items that appeared difficult to recall (as indicated by the Rasch performance estimate) yet were confidently recalled, appeared as more satisfying than an equally confident recall of an item that was easier to retrieve. We therefore tested the model Satisfaction ~ Confidence x Rasch performance + Rasch performance | participant. Indeed, the interaction was reliable negative at bConfidence x Rasch = -.25, t = -2.83, p < .01, the likelihood ratio for the interaction model over the null model was LR = 1266, p < .01 suggesting that for equal confidence, more difficult items (i.e., low Rasch performance) induced higher satisfaction ratings when successfully recalled.” Here the authors report a significant Confidence × Rasch performance interaction and describe what it means. However, the interaction model (Equation 4) is compared against a null model without the interaction term and without fixed-effect terms. Our contention is that to assess the role of the interaction, a full model (Equation 6) should be compared with a simpler model with the other fixed-effect terms and without the interaction term (Equation 5).

5. Satisfaction ~ Confidence + Rasch performance + (Rasch performance | participant)

6. Satisfaction ~ Confidence + Rasch performance + Confidence × Rasch performance + (Rasch performance | participant)

If this model comparison proves significant, then the Results, Discussion and Abstract should emphasize the interaction, as it is potentially a novel and interesting finding. Unlike argued by the authors in the Discussion, lines 364-373, successful recall of more difficult items, for a given confidence level, would be associated with higher levels of satisfaction (a result consistent with the “desirable difficulties” framework).

9. Model comparisons (part 2). The excerpts below refer to lines 319–330 in the manuscript. In this passage of the Results, the authors also describe a series of mixed-effect model comparisons. The analyses here (part 2) focus on the fixed-effect terms log(time) (time since encoding) and Session performance (participant session performance average recall). For ease of reference, we will use numbered equations to refer to each mixed-effect linear regression model and the text passage it refers to.

(1) “Furthermore, we employed a mixed effects linear regression test of retrieval satisfaction for correctly recalled items as a function of log time since encoding as fixed effect and with participant as random variable. With 1172 observations, the fixed effect of log time since encoding was blog(time) = -0.027, t = -2.19, p < .05, with LR = 9.19, p < .001.”

1. Satisfaction ~ log(time) + (1 | participant)

(2) “The corresponding analysis with time since encoding as a predictor of confidence rating yielded blog(time) = -0.03, t = -2.73, p < .01 with LR = 6.30, p < .05.”

2. Confidence ~ log(time) + (1 | participant)

(3) “Finally, noticing that recall rates varied substantially across sessions and between participants (see left panel of figure 3), we tested participant session performance average recall score as a predictor of correct recall satisfaction and response confidence, respectively. For satisfaction, we found that session performance bsession perf. = 0.50, t = 3.35, p < .001 and LR = 9.19, p <.01.”

3. Satisfaction ~ Session performance + (1 | participant)

(4) “For confidence ratings, session average performance bsession perf. = 0.59, t = 3.88, p < .0001 and LR = 12.0, p < .0001.”

4. Confidence ~ Session performance + (1 | participant)

Here we ask why the authors did not follow a sequence of model comparisons similar to the ones carried out for Confidence and Rasch performance (i.e., model comparisons with progressively more complex models). In particular, we ask if the authors tested whether Satisfaction is affected by a log(time) × Rasch performance interaction. This interaction would be of considerable theoretical interest because both log(time) and Rasch performance are terms related to item difficulty. It thus stands to reason that the Satisfaction for the most difficult combination would be very different from Satisfaction for the least difficult combination (most difficult combination: high log(time) and low Rasch performance; least difficult combination: low log(time) and high Rasch performance). A model comparison between models 5 and 6 below would address this question.

5. Satisfaction ~ log(time) + Rasch performance + (1 | participant)

6. Satisfaction ~ log(time) + Rasch performance + log(time) × Rasch performance + (1 | participant)

10. Please include on the OSF site the analyses’ files for the Rasch models and the mixed-effects linear regressions. Rasch models are not common in the memory literature, and so it would be very helpful for other researchers to have access to an implementation of this type of analysis. In addition, mixed-effect analyses may become complicated to follow from text only (as opposed to looking at the analysis code). Thus, we also ask the authors to make the mixed-effects linear regression code available.

Minor points

-Line 141: “Out of 70 participants who completed the first session a total of 30 participants (…) completed all sessions…”. Attrition rate was very high (> 50%). At a minimum, the authors should present a summary of the main characteristics of this dropout sample (mean age, dropout pattern – e.g., most dropped after the first session, followed by second session and son on?). This could provide a more complete picture for the reader as to what kind of participant remained in the studied (e.g., unusually motivated students with higher levels of intrinsic curiosity). For example, one could argued that the relationship between satisfaction and confidence observed in the study, may suffer from self-selection bias, in that individuals who already find the act of recall rewarding (satisfying) were the one that remained in the final sample of the study.

-Some abbreviations are not in their long form upon first presentation. Examples: Line 171 - GDPR (General Data Protection Regulations), Line 284 - LR (Likelihood ratio)

-The terms “word list” (line 148) vs. “wordlist” (line 178) are used interchangeably in the text. We suggest the use of a single form (e.g., “word list”).

-Line 204: Encoding session the 7th day => Encoding session in the 7th day

-Lines 188–192: Small inconsistencies in the response scales. Most scales present a prefix “Un-” on their lowest value, but there is no prefix for “Satisfied” (i.e., “Not Satisfied”). Was there any reason for this choice of words? One possibility would have been “Dissatisfied”, as in (-1 = Dissatisfied, +1 = Satisfied) – same applies to lines 211-212 (-1 = Not Satisfied, +1 = Satisfied).

-Lines 188–192: It was not immediately clear how many levels each slide bar entails (e.g., three values as in [-1, 0 , 1] or five values as in [-1 , -.5, 0, +5, +1]). From the text, I implied that the slide scale is continuous, with values ranging from -1 to 1. It would help to make this more explicit in the text.

-Line 195: “…similarities according to the Levenshtein’s distance algorithm”. Since the use of the Levenshtein’s algorithm is not very common in memory studies, it would be helpful to add a sentence or two giving a short explanation of the algorithm (i.e., how it considers as correct a response in which a typo was mistakenly typed by the participant).

Lines 278: “recall and confidence may interact in satisfaction” => may interact with satisfaction

Reviewer #2: Comments relevant to question 1 above:

I understand the notion of intrinsic reward and the use of imaging to document activation of neural reward circuitry during certain memory tasks; some relevant imaging work is cited in this paper. However, I am having trouble with the particular behavioral method used to measure reward in the current work. The primary variable used as a proxy for reward is the participant's satisfaction with each response. To me, asking how confident you are with your answer seems synonymous with asking how satisfied you are with your answer, and indeed, you obtained very high correlations between the two. I don't necessary think that being satisfied with an answer means that you feel reward; it just means you think you got it right. My worry is that your results don't tell us that people feel rewarded by accurate retrieval but simply that people are pretty good at judging their accuracy. The interaction between confidence, item difficulty, and satisfaction was the only place where I could start to see these as different. Line 384 suggests that there may be additional outcomes that would demonstrate the validity of how satisfication and confidence were measured and how well the construct of satisfaction maps onto reward. Can you integrate these into the paper? Also, what were the specific instructions given to the participants re these measures? Did you define what you meant by being satisfied as distinct to what you meant by being confident? And finally, can you clarify and expand upon what you mean in line 99-100 "the presence of satisfaction following a reward constitutes a sufficient criterion for the reward"?

Your sample is mostly female. Is there any literature to suggest differences in the interface between memory and reward by sex or gender? If so, this should be included. Should the skewed sample be acknowledged as a limitation of this study?

On line 195, a sentence explaining the Levenshtein's distance algorithm is needed.

You measured confidence re remembering correctly and confidence re the likelihood of remembering one week later but these distinctions are not represented in the results.

In several places you present means but no standard deviations or any indication of the range of outcomes.

Comments related to #4 above:

line 32 reads as if there were five word pairs, not five study sessions with 100 word pairs

line 49 should be 'is it possible'

line 126 should be 'phenomena' also specify more common than WHAT?

lines 175 and 181, should this be 'blinking' or 'blank'?

line 277 should be 'exists'

line 298 = sentence fragment

6. PLOS authors have the option to publish the peer review history of their article (what does this mean?). If published, this will include your full peer review and any attached files.

Reviewer #1: **Yes: **Luciano Grüdtner Buratto

Reviewer #2: No

---

## [Author Response · Author response to Decision Letter 0]

28 Aug 2023

Point by point response to reviewers’ comments and questions.

R1:

1. In the title and throughout the manuscript, the use of the term reliable and its variants (e.g., “reliable retrieval”) seemed misleading to me. I understand that reliability, in memory research, refers to the accuracy or the “truthfulness” of the recalled information according to some external objective criterion (e.g., in eyewitness testimony research). Reliability may also have very specific meanings in psychology, such as test-retest reliability. However, in this manuscript, the term reliability was used as synonymous with confidence (e.g., “recall confidence”), which refers to the degree of subjective certainty learners assign to their response (i.e., the subjective confidence in their own memory). The interchangeable use of the terms reliability and confidence may inadvertently confuse the reader. To improve readability and keep consistency with the memory literature in general, I suggest replacing the term reliability and its variants with confidence and its variants (e.g., confident retrieval or memory confidence) when the aim is to refer to learners’ subjective judgments. 

Thanks for pointing out the important distinction. We understand the that the original paper may have introduced confusion on this note. However, we need both terms (reliability in the objective sense and confidence in the subjective sense, respectively) and their respective meanings to describe the purpose and design of our study. Humans only have access to indirect measurements via imperfect observations which render all empirical knowledge subject to some level of confidence. It follows that the brain too only has indirect assessments available to reinforce behavior and we find that confidence is a reasonable candidate signal. But importantly, and for adaptive fitness, the confidence and reward-policy must be tied to objective states. We have revised the paper with a view towards clarifying this distinction. 

2. In the Abstract, the authors make the following statement (brackets added by me): “We found that retrieval satisfaction decreased with time since encoding and [also decreased with] session retrieval performance.” This statement seems to be based on results presented (lines 281–285), showing a Confidence × Rasch Performance interaction. However, in the Results (lines 328–329), the authors claim that “… For satisfaction, we found that session performance bsession perf. = 0.50, t = 3.35, p < .001 and LR = 9.19, p <.01.” The term “session [retrieval] performance” seems to imply two different meanings throughout the manuscript. In Rasch analyses, it seems to refer to the estimated item-by-participant performance in the first retrieval attempt across the encoding sessions (based on items and participants parameters estimated with the Rasch model), a proxy of pre-retrieval uncertainty, as stated by the authors (one estimate per participant). In final-recall test analyses, however, “session performance” seems to refer to the observed final-recall performance (by session; five estimates per participant), used to assess whether the empirical final recall as a function of different retention intervals affect retrieval satisfaction and retrieval confidence. These two senses of “session performance” seem to show different relationships with satisfaction. To improve readability, I strongly recommend that the authors make this distinction 

explicit and, in the Discussion section, discuss the implications of each of these results for the hypotheses presented. 

Thanks, and we apologize for not making this clearer. The entire manuscript has been revised with an eye on both clarifying this confusion and using better terms consistently. We now use the term “study session memory” to refer to the proportion of items correctly recalled from each study session (thus five values / participant) and “Rasch performance” to indicate the individual x item - derived probability of answering correctly to a specific item. 

3. Lines 217–220: “Upon pressing yes, the user was presented with text explaining a time gap before the answer was presented. After a 5 s waiting time elapsed, the Swahili word that they wished to know the English translation to was presented on the left, and the translation on the right of a crosshair in the center of the screen”. 

- Why did the authors insert a 5-s delay until the presentation of the correct English word? This introduces a “cost of response”, rendering the response “Yes, I am curious” more time consuming than “No, I am not curious". Given the context of a long, multisession experiment, in which participants are already tired, introducing such 5-s cost is likely to reduce the number of “Yes” responses. Therefore, we suggest that the authors explain why they have chosen to include the 5-s delay and how it could have affected the results and conclusions of their study. 

We apologize for not making this clearer in the original submission. Willingness to wait is a common method to quantify subjective value – and is frequently used to assess curiosity as a behavior (one measured quantity is sacrifized in favor of satisfying curiosity – the investment quantifies curiosity). We have now included clarifying text in the final paragraph of the introduction. We do not see any threat to the validity of the results based on the feedback waiting procedure – e.g., confidence and satisfaction ratings were made before waiting and should therefore not have been affected by intermittent waiting periods.

4. Lines 231–232: “Participants recalled 61% of the items in their first recall attempt and never needed more than five repetitions until they correctly recalled an item”. One characteristic of the adopted within-subjects design is that it confounds the time since encoding with the number of intervening lists until the final test (which, speculatively, may have led to a buildup of proactive interference). Did the initial recall reported in the text differ across the five sessions? Please explicitly state this information in the text. 

Just to clarify – the 61% initial performance is with respect to first exposure within a list. The point raised here concerns the potential for interference between lists (sessions). The expectation then would be slightly worse performance in the final recall test for items studied in the latter sessions due to interference. 

Performance was not equal across study-sessions in terms of number of repetitions. Specifically, in the first session, participants needed more repetitions to reach criterion than in the latter sessions resembling task adaptation. Presumably, this may have had an impact on the forgetting rate too. However, if so, it still does not substantially affect the validity of our results and interpretations thereof, as the main point is that we imposed a retention difference by spacing study sessions and found an effect in terms of recall rates, in the aggregate. More importantly, participants varied in the amount they recalled from different sessions in the final recall test and that variability was highly related to confidence and satisfaction ratings, respectively. Finally, the difference in initial performance would have been a potential problem had we not employed the repetition method which guaranteed evidence of learning of all words by all participants across all sessions. We have now reported amount of repetitions required across the sessions and provide a statistical analysis.

5. Lines 232–235: The authors compared satisfaction and confidence between correct and incorrect retrieval attempts. Among the incorrect attempts, what was the rate of response omissions? If the omission rate was high, I believe it is important to restrict the analyses only to incorrect attempts with responses, as it would not be surprising for satisfaction and confidence to be higher in correct attempts if the incorrect attempts include many omissions. 

By our analysis we wanted to point out that a major source of variance in satisfaction and confidence – as the reviewer state – is attributed to self-identified performance. We agree with R1 that comparing satisfaction and confidence ratings restricted to incorrect responses with entries (i.e., eliminating omissions) may constitute a cleaner analysis in the sense that incorrect responses do not a priori need to be identified as incorrect. We now include both analyses (i.e., with(out) omissions) in the revised paper. Moreover and in the spirit of consistency, we carried out the corresponding analyses with respect to final recall test performance. Across all four comparisons we received the same pattern of statistically reliable results, namely that correct responses were rated higher both in terms of satisfaction and confidence than incorrect responses.

6. Lines 247–248: “As seen in figure 1, participants’ retrieval confidence and satisfaction increased between first and final retrieval attempt”. Given the dropout procedure adopted in the study, I wonder how the authors defined “final retrieval attempt.” For example, if for a given participant, the last test repetition led to correct recall of the remaining five items, did only those five items contribute to the measures of confidence and satisfaction? Or, alternatively, did the final attempt of each item contribute to the measures of confidence and satisfaction? Could you explain this point? 

To make the ratings comparable across items and sessions, we reported only the first (in which recall accuracy varied) ratings and final ratings (which were by definition always correct) in the results. Please note that the final ratings were identical to their first ratings if the first recall attempt was correct. But also note, that we only use the final recall ratings to compute learning-related change here (figure 1). The association between confidence and satisfaction ratings were based on the first recall attempt data. We have now clarified this in the results text.

7. The sample size (n = 30) is quite small for estimating Rasch models. Some points: 

a) Please discuss the consequences of the small sample size on the Rasch model estimates (e.g., how would that impact on the relationship between item difficulty and satisfaction)? 

It is true the sample size was rather small for this type of analysis. Despite this, the Rasch estimates rather faithfully reproduced participant- and item-based performance averages. We have now reported these results in the results section. For further information about the outcome of the Rasch analysis, we have now posted the analysis code (matlab) on OSF in which a rather extensive analysis may be found.

b) Rasch model estimates were used as indices of item difficulty. Nelson and Dunlosky (1994) provide norms of recall difficulty for a sample of 100 Swahili-English word pairs. Why did the authors decide to estimate item difficulty from Rasch models rather than from Nelson and Dunlosky’s norms? As a suggestion, the authors could correlate their Rasch model estimates with Nelson and Dunlosky’s (1994) norm scores as a means of validating the items’ difficulty estimates. 

We used a different set of words compared to Nelson and Dunlosky (1994) and can therefore not compare our results. Psychometrically, distributions in terms of performance are rather normal distributed. In addition to Nelson and Dunlosky, we aimed at restricting word lengths of the English word to make the grading via Levenshtein edit distance more congruent across items; a longer word (in Nelson and Dunlosky up to 8 letters) allow for more spelling errors. Rasch model estimates also have the appeal of being based on the empirical set of test items used. As R1 says, item response analyses is not that common in cognitive psychology literature despite its relevance for interpretation of results. Using it here both serves the purpose of providing better insight into the psychometric quality of our test items, and also offers an estimate of participants estimated probability of responding correctly. 

c) Additionally, the authors could correlate item difficulty estimates from Figure 2 to the proportion recalled in the final test from Figure 3 (left panel) as an attempt to validate independently their estimates of the Rasch models.

This is a good idea. Just bear in mind that the Rasch analysis is based on first recall attempt performance in the study sessions. Moreover, the final recall test performance additionally is subject to the impact of varying retention delays. That said, the association between initial Rasch estimates and final recall performance were r2 = .30, p < .01. Indeed, the Rasch estimates also turned out to capture aspects of satisfaction also in the final recall test (see below).

 8. Model comparisons (part 1). The excerpts below refer to lines 267–285. In this section of the Results, the authors describe a series of mixed-effect model comparisons to support the patterns described in the figures. The analyses here (part 1) focus on the fixed-effect terms Rasch performance (item difficulty) and Confidence and their interaction. For ease of reference, we will use numbered equations to refer to each mixed-effect linear regression model and the text passage it refers to. 

The main question we have here is whether the full model, with the fixed effects of Confidence, Rasch performance and their interaction (Equation 6) is significant when compared to simpler models (Equations 3, 4 and 5). If it is, the role of the interaction should be emphasized in the Abstract and in the Discussion, as the result implies that that Satisfaction is influenced by response Confidence, item difficulty and the Confidence × Rasch performance interaction (for a given level of confidence, correct responses to more difficult items are associated with higher satisfaction). 

Below we reproduce the models and the corresponding text to elaborate on our issue regarding model comparison. The simplest model assessed was the following: 

1. Satisfaction ~ Rasch performance + (Rasch performance | participant) 

(1) “The model Satisfaction ~ Rasch performance + Rasch performance | participant yielded bRasch = 0.35, t = 2.73, p < .01 with 1820 observations. The model with Rasch performance as a fixed effects term was reliably better supported than a null model without the fixed effects term, LRfixed = 8.0, p < .01.” Thus, participants’ satisfaction with their own answers was predicted by item difficulty. The model with the Rasch performance term was compared to the model without that term. The authors also assessed whether confidence could be predicted from Rasch performance. It did: 

2. Confidence ~ Rasch performance + (Rasch performance | participant) 

(2) “We tested the corresponding effect replacing satisfaction with confidence and found that bRasch = .40, t = 3.2, p < .01. The model with Rasch performance as fixed effects term was reliably better supported than a null model without the term, LR = 10.0, p < .001.” Next, the authors assessed Satisfaction as function of Confidence: 

3. Satisfaction ~ Confidence + (Rasch performance | participant) 

(3) “We also tested response confidence as a predictor of satisfaction by running confidence ratings as a fixed effects term in a linear mixed effects model and found a reliable association between satisfaction and confidence ratings with b1 = 0.72, t = 17.0, p < .001. The results indicated that the easier the item is to recall, the more satisfying is the experience.” Here, the last sentence refers to item difficulty, when it should refer to Confidence, as Confidence was the fixed-effect term assessed in Equation 3. The authors then proceed to assess whether the interaction term is significant: 

 Agreed, Satisfaction ~ Confidence is not about objective difficulty but subjective. Language in the results section has been revised accordingly.

4. Satisfaction ~ Confidence × Rasch performance + (Rasch performance | participant) 

(4) “However, the possibility still exist that ease recall and confidence may interact in satisfaction such that items that appeared difficult to recall (as indicated by the Rasch performance estimate) yet were confidently recalled, appeared as more satisfying than an equally confident recall of an item that was easier to retrieve. We therefore tested the model Satisfaction ~ Confidence x Rasch performance + Rasch performance | participant. Indeed, the interaction was reliable negative at bConfidence x Rasch = -.25, t = -2.83, p < .01, the likelihood ratio for the interaction model over the null model was LR = 1266, p < .01 suggesting that for equal confidence, more difficult items (i.e., low Rasch performance) induced higher satisfaction ratings when successfully recalled.” Here the authors report a significant Confidence × Rasch performance interaction and describe what it means. However, the interaction model (Equation 4) is compared against a null model without the interaction term and without fixed-effect terms. Our contention is that to assess the role of the interaction, a full model (Equation 6) should be compared with a simpler model with the other fixed-effect terms and without the interaction term (Equation 5). 

5. Satisfaction ~ Confidence + Rasch performance + (Rasch performance | participant) 

6. Satisfaction ~ Confidence + Rasch performance + Confidence × Rasch performance + (Rasch performance | participant) 

If this model comparison proves significant, then the Results, Discussion and Abstract should emphasize the interaction, as it is potentially a novel and interesting finding. Unlike argued by the authors in the Discussion, lines 364-373, successful recall of more difficult items, for a given confidence level, would be associated with higher levels of satisfaction (a result consistent with the “desirable difficulties” framework). 

We tested the suggested analyses and did indeed receive reliable support for it! Equation 6 in R1’s numbering does a better job of accounting for the results than equation 5, indicating that there is indeed an interaction such that for equal confidence, correctly recalling a more difficult item is more satisfying. The results have been revised and now contains the proposed analyses and results. Please note that the interaction effect is modest in relation to the main effect of difficulty (and confidence) accounting for satisfaction. Nonetheless, the result of this new analysis warrants a slight revision also of our conclusions.

9. Model comparisons (part 2). The excerpts below refer to lines 319–330 in the manuscript. In this passage of the Results, the authors also describe a series of mixed-effect model comparisons. The analyses here (part 2) focus on the fixed-effect terms log(time) (time since encoding) and Session performance (participant session performance average recall). For ease of reference, we will use numbered equations to refer to each mixed-effect linear regression model and the text passage it refers to. 

(1) “Furthermore, we employed a mixed effects linear regression test of retrieval satisfaction for correctly recalled items as a function of log time since encoding as fixed effect and with participant as random variable. With 1172 observations, the fixed effect of log time since encoding was blog(time) = -0.027, t = -2.19, p < .05, with LR = 9.19, p < .001.” 

1. Satisfaction ~ log(time) + (1 | participant) 

(2) “The corresponding analysis with time since encoding as a predictor of confidence rating yielded blog(time) = -0.03, t = -2.73, p < .01 with LR = 6.30, p < .05.” 

2. Confidence ~ log(time) + (1 | participant) 

(3) “Finally, noticing that recall rates varied substantially across sessions and between participants (see left panel of figure 3), we tested participant session performance average recall score as a predictor of correct recall satisfaction and response confidence, respectively. For satisfaction, we found that session performance bsession perf. = 0.50, t = 3.35, p < .001 and LR = 9.19, p <.01.” 

3. Satisfaction ~ Session performance + (1 | participant) 

(4) “For confidence ratings, session average performance bsession perf. = 0.59, t = 3.88, p < .0001 and LR = 12.0, p < .0001.” 

4. Confidence ~ Session performance + (1 | participant) 

Here we ask why the authors did not follow a sequence of model comparisons similar to the ones carried out for Confidence and Rasch performance (i.e., model comparisons with progressively more complex models). In particular, we ask if the authors tested whether Satisfaction is affected by a log(time) × Rasch performance interaction. This interaction would be of considerable theoretical interest because both log(time) and Rasch performance are terms related to item difficulty. It thus stands to reason that the Satisfaction for the most difficult combination would be very different from Satisfaction for the least difficult combination (most difficult combination: high log(time) and low Rasch performance; least difficult combination: low log(time) and high Rasch performance). A model comparison between models 5 and 6 below would address this question. 

5. Satisfaction ~ log(time) + Rasch performance + (1 | participant) 

6. Satisfaction ~ log(time) + Rasch performance + log(time) × Rasch performance + (1 | participant) 

While the proposed method has the merit of testing the relationship between experimentally induced difficulties, it seems theoretically unclear why there should be an interaction between the two – after all, they are just two different and independent sources of difficulty. 

Instead, it seems the more theoretically relevant question is how confidence might interact with difficulty in driving retrieval satisfaction as suggested by R1 in the previous comment section. We therefore separately tested the interactions between confidence and Rasch performance, and confidence and retention time, respectively, employing the null models suggested by R1. We found support for an inverse relationship between confidence and Rasch performance replicating the finding from the study sessions. The results indicated that more difficult-to-encode items that are confidently recalled yield higher satisfaction than confident recall of easier-to-encode items. We did however not find any reliable interaction between retention time and confidence. The new analyses have been added to the results section. Furthermore, we bring up the modest effect of retention time and non-significant interaction with confidence in the discussion section. 

10. Please include on the OSF site the analyses’ files for the Rasch models and the mixed-effects linear regressions. Rasch models are not common in the memory literature, and so it would be very helpful for other researchers to have access to an implementation of this type of analysis. In addition, mixed-effect analyses may become complicated to follow from text only (as opposed to looking at the analysis code). Thus, we also ask the authors to make the mixed-effects linear regression code available. 

The analysis code is uploaded to the OSF. To run the matlab code you also need the IRTm package provided from https://ppw.kuleuven.be/okp/software/irtm/

Information about this package dependence is provided at the OSF wiki for the retrieval satisfaction project.

Minor points 

-Line 141: “Out of 70 participants who completed the first session a total of 30 participants (…) completed all sessions…”. Attrition rate was very high (> 50%). At a minimum, the authors should present a summary of the main characteristics of this dropout sample (mean age, dropout pattern – e.g., most dropped after the first session, followed by second session and son on?). This could provide a more complete picture for the reader as to what kind of participant remained in the studied (e.g., unusually motivated students with higher levels of intrinsic curiosity). For example, one could argued that the relationship between satisfaction and confidence observed in the study, may suffer from self-selection bias, in that individuals who already find the act of recall rewarding (satisfying) were the one that remained in the final sample of the study. 

Thanks for the comment. We have now included a section with subheading “Attrition analysis” in the results that covers attrition. We find a slight bias for younger and females to be more likely to complete the study but the core relationship between satisfaction and confidence stands also when tested with data only from the first study session.

-Some abbreviations are not in their long form upon first presentation. Examples: Line 171 - GDPR (General Data Protection Regulations), Line 284 - LR (Likelihood ratio) 

Thanks for spotting this – we have now revised the manuscript accordingly.

-The terms “word list” (line 148) vs. “wordlist” (line 178) are used interchangeably in the text. We suggest the use of a single form (e.g., “word list”). 

Now changed to word list

-Line 204: Encoding session the 7th day => Encoding session in the 7th day 

done

-Lines 188–192: Small inconsistencies in the response scales. Most scales present a prefix “Un-” on their lowest value, but there is no prefix for “Satisfied” (i.e., “Not Satisfied”). Was there any reason for this choice of words? One possibility would have been “Dissatisfied”, as in (-1 = Dissatisfied, +1 = Satisfied) – same applies to lines 211-212 (-1 = Not Satisfied, +1 = Satisfied). 

We did discuss choice of phrasing quite extensively in the lab. One lab member with American English as first language and another member with British English as first language suggested “Not satisfied” to be more appropriate. Looking at response distributions, the inconsistency does not seem to translate to a difference in terms of useage compared to the other scales (e.g., more dichotomized responses).

-Lines 188–192: It was not immediately clear how many levels each slide bar entails (e.g., three values as in [-1, 0 , 1] or five values as in [-1 , -.5, 0, +5, +1]). From the text, I implied that the slide scale is continuous, with values ranging from -1 to 1. It would help to make this more explicit in the text. 

Rating scales were continuous. Thanks for the comment. We have now revised and clarified this in the methods section.

-Line 195: “…similarities according to the Levenshtein’s distance algorithm”. Since the use of the Levenshtein’s algorithm is not very common in memory studies, it would be helpful to add a sentence or two giving a short explanation of the algorithm (i.e., how it considers as correct a response in which a typo was mistakenly typed by the participant). 

We have now briefly explained the Levenshtein distance. See lines 217-220.

Lines 278: “recall and confidence may interact in satisfaction” => may interact with satisfaction

Done

R2

The primary variable used as a proxy for reward is the participant's satisfaction with each response. To me, asking how confident you are with your answer seems synonymous with asking how satisfied you are with your answer, and indeed, you obtained very high correlations between the two. I don't necessary think that being satisfied with an answer means that you feel reward; it just means you think you got it right. My worry is that your results don't tell us that people feel rewarded by accurate retrieval but simply that people are pretty good at judging their accuracy

The surprising aspect here is that satisfaction and confidence appear as synonymous but need not be. Confidence itself in cognition does not in and of itself imply any emotional or reward aspect. We derived two possible relationships from literature essentially with conflicting predictions and tested those. We found rather strong evidence for one of those predictions. This relationship has not been reported much before in the cognitive science literature to our knowledge (the Clos et al. reference being the only close tie). We attempt and to some extent show that confidence and satisfaction are slightly dissociated via interactions between e.g., item difficulty and confidence (please see below and in our comments to R1 as well as the revisions in the paper). What we think is the interesting contribution here is the possibilities that open up as we think about confidence as a core signal for motivated cognition.

The interaction between confidence, item difficulty, and satisfaction was the only place where I could start to see these as different. Line 384 suggests that there may be additional outcomes that would demonstrate the validity of how satisfication and confidence were measured and how well the construct of satisfaction maps onto reward. Can you integrate these into the paper?

Please see responses to R1 on the note of interaction. Indeed we did find more of it and the paper has been revised accordingly.

Also, what were the specific instructions given to the participants re these measures? Did you define what you meant by being satisfied as distinct to what you meant by being confident?

There were no additional instructions beyond the rating requests to dissociate the two concepts as we were reluctant to also inadvertently influence interpretations. 

And finally, can you clarify and expand upon what you mean in line 99-100 "the presence of satisfaction following a reward constitutes a sufficient criterion for the reward"?

We apologize for being cryptic. The sentence has been revised and the section extended to clarify our point.

Your sample is mostly female. Is there any literature to suggest differences in the interface between memory and reward by sex or gender? If so, this should be included. Should the skewed sample be acknowledged as a limitation of this study?

We know of no such literature. We agree the sample is highly skewed and that is obviously problematic if the study motive strongly claimed population wide generalizability. As of now, the core aim was to demonstrate the effect – which we did. That said, it might be quite interesting to pursue individual differences in terms of how their cognitive satisfaction and reward policies may differ.

On line 195, a sentence explaining the Levenshtein's distance algorithm is needed.

Done

You measured confidence re remembering correctly and confidence re the likelihood of remembering one week later but these distinctions are not represented in the results.

Thanks for pointing this out and we apologize for the confusion. The question about remembering one week into the future is a judgment of learning (JOL) whereas the other question about confidence in retrieval relates to the current sensation when the word was (or was not) recalled. We have now revised the methods section to clarify this point. Also, ss seen in the results section, the JOL allow us to further characterize the meta-mnemonic competence of the participants in the study by demonstrating that their last JOL from the study sessions predict performance in the final recall test.

In several places you present means but no standard deviations or any indication of the range of outcomes.

The text has now been revised and corrected where only means were previously presented. Distributions in the mixed models are challenging to display well either in text, tables or graphs. In striking a balance between proximity to original data and readability, we offer individual linear regression fits in figures 2 and individual participants ‘session performance means in figure 3. We encourage a look at the data file and the analysis code at the OSF depository for more in-depth appreciation of the data.

line 32 reads as if there were five word pairs, not five study sessions with 100 word pairs

Thanks – we have now revised that sentence.

line 49 should be 'is it possible'

Agreed – revised accordingly.

line 126 should be 'phenomena' also specify more common than WHAT?

Apologies for the oversight – the sentence is now complete.

lines 175 and 181, should this be 'blinking' or 'blank'?

thanks for the note - revised

line 277 should be 'exists'

done

line 298 = sentence fragment

done

---

## [Decision Letter · Decision Letter 1]

18 Sep 2023

PONE-D-23-11730R1Reliable retrieval is intrinsically rewarding:

Recency, item difficulty, study session memory, and subjective confidence predict satisfaction in word-pair recallPLOS ONE

Dear Dr. Holm,

Thank you for submitting your manuscript to PLOS ONE. After careful consideration, we feel that it has merit but does not fully meet PLOS ONE’s publication criteria as it currently stands. Therefore, we invite you to submit a revised version of the manuscript that addresses the points raised during the review process.

We look forward to receiving your revised manuscript.

Kind regards,

Stergios Makris

Academic Editor

PLOS ONE

Journal Requirements:

Reviewers' comments:

Reviewer's Responses to Questions

**Comments to the Author**

1. If the authors have adequately addressed your comments raised in a previous round of review and you feel that this manuscript is now acceptable for publication, you may indicate that here to bypass the “Comments to the Author” section, enter your conflict of interest statement in the “Confidential to Editor” section, and submit your "Accept" recommendation.

Reviewer #1: (No Response)

Reviewer #2: All comments have been addressed

2. Is the manuscript technically sound, and do the data support the conclusions?

Reviewer #1: Yes

Reviewer #2: (No Response)

3. Has the statistical analysis been performed appropriately and rigorously? 

Reviewer #1: Yes

Reviewer #2: (No Response)

4. Have the authors made all data underlying the findings in their manuscript fully available?

Reviewer #1: Yes

Reviewer #2: (No Response)

5. Is the manuscript presented in an intelligible fashion and written in standard English?

Reviewer #1: Yes

Reviewer #2: (No Response)

6. Review Comments to the Author

Reviewer #1: PONE-D-23-11730 – Review R1

Summary

The authors have addressed satisfactorily the issues raised in the previous version. Here I would like to highlight minor points concerning clarity and copyediting details before recommending the manuscript for publication.

Clarity

1. In the Figure 1, panels A and B, it is not entirely clear to me what the x- and y-axis represent. In the attached file is a copy of Fig. 1(A) where two data points have been highlighted. My understanding is as follows. Each point represents one participant. The participant in the green square represents someone who had a relatively good initial recall in the first session (e.g., mean correct recall in session 1 = .51) and a perfect recall in the least session (mean correct recall in session 5 = 1.0; I presumed this latter case applied to all cases, since the design followed a drop-out procedure). Thus, the retrieval improvement on the x-axis for the participant in green is 1.0 − .51 = .49. The learning-related confidence change is the difference between the mean confidence for the correct attempts in session 1 and the mean confidence for the correct attempts in session 5.

Following the same logic, the retrieval improvement for the participant in the red box is the mean correct recall in session 5 (1.0) minus the mean correct recall in session 1 (e.g., 0.10). Thus, the retrieval improvement for the participant in the red box is high at .90. The corresponding difference in confidence ratings is low at 0.09. Is that reasoning correct? It would be helpful if the authors described in a short paragraph how they obtained one of the points in Fig. 1, panel A.

Minor points

In the following, I present a list of small suggestions to improve readability and compliance with Plos One copyediting standards.

-p.2, line 30: “to successfully retrieve” → “to successfully retrieve it”.

-p.2, line 36-37: “We found that retrieval satisfaction decreased with time since encoding and study session retrieval performance”. → “We found that retrieval satisfaction decreased with time since encoding and with lower study session retrieval performance”.

-p.2, line 38: “But we also found … indicating” → “We also found … , indicating”.

-p.5, line 105: “reward at play the presence of” → “reward at play in the presence”.

-p.11, line 255: The data file in the OSF platform was uploaded with the .txt extension. It would be more helpful if it were uploaded with the the .csv extension, as the OSF system would be able to read the file and print it on the screen of the web browser.

-p.12, line 289 (section “Encoding section results” and wherever applicable): The abbreviations for mean (M) and standard deviation (SD) should be reported in capital letters and italics. Please revise all such cases in the manuscript (e.g., p.15, section “Final recall test results”).

-p.13, line 309 (Figure 1 caption): The first letters should be capitalized in the boldface caption: “Study Session Learning and Ratings”. In addition, the legends in the x- and y-axis should have the first letter of each word capitalized: x = Retrieval Improvement; y = Learning-related confidence change. Same applies for panels B and C.

-p.13, line 317: References to Figures should be abbreviated (Figure 1 → Fig. 1; apply throughout the text for figure 1 → Fig. 1, figure 2 → Fig. 2, and figure 3 → Fig. 3).

-p.15, line 362 (Figure 2 caption): The first letters are capitalized in the boldface caption: “Performance and Rating Associations”. The legends in the x- and y-axis should have the first letter of each word capitalized: x = Estimated performance (Rasch); y = z(Satisfaction Rating). Same applies for panels B and C.

-p.17, line 413 and line 415: Change the b subscripts to keep consistency with new terminology (performance memory → session memory), so that b_{session.perf} becomes b_{session.mem}. One case on p. 413, and the other on p. 415.

-p.18, line 436: “z-scored JOL rating at study.” → “z-scored JOL rating unit at study.”

-p.20, line 489: “…participants who did not complete the experiment were slightly more likely to be male …”. Shouldn’t it be female? On p. 11, line 262, the authors say that “Of those who did not complete all sessions, 20 identified as female and 11 as male”. So, the claim on p.20, line 489 should be “more likely to be female”.

-p.20, line 499: “…, what then might the adaptive value … be?” → “…, what then might be the adaptive value …?”

-p.22, line 542: Here I am not sure which preposition is the most appropriate, as it depends on the meaning the authors wish to convey. The text reads “It is also possible that the learning reward argument in curiosity theory is incorrect or at least that the hedonic ‘wanting’ (Berridge & Kringlebcach, 2015) [Note: please use Plos One reference style here] aspect of reward is irrelevant in curiosity.” Shouldn’t it be “irrelevant to curiosity”?

Reviewer #2: (No Response)

7. PLOS authors have the option to publish the peer review history of their article (what does this mean?). If published, this will include your full peer review and any attached files.

Reviewer #1: **Yes: **Luciano Grüdtner Buratto

Reviewer #2: No

---

## [Author Response · Author response to Decision Letter 1]

26 Sep 2023

Rebuttal 2

Please find below our responses to the Reviewer’s comments. The reviewer’s comments are italics, our responses are in plain text.

In the Figure 1, panels A and B, it is not entirely clear to me what the x- and y-axis represent. In the attached file is a copy of Fig. 1(A) where two data points have been highlighted. My understanding is as follows. Each point represents one participant. The participant in the green square represents someone who had a relatively good initial recall in the first session (e.g., mean correct recall in session 1 = .51) and a perfect recall in the least session (mean correct recall in session 5 = 1.0; I presumed this latter case applied to all cases, since the design followed a drop-out procedure). Thus, the retrieval improvement on the x-axis for the participant in green is 1.0 − .51 = .49. The learning-related confidence change is the difference between the mean confidence for the correct attempts in session 1 and the mean confidence for the correct attempts in session 5.

Your interpretation is correct. The caption has been revised to clarify the meaning of the scatter.

Following the same logic, the retrieval improvement for the participant in the red box is the mean correct recall in session 5 (1.0) minus the mean correct recall in session 1 (e.g., 0.10). Thus, the retrieval improvement for the participant in the red box is high at .90. The corresponding difference in confidence ratings is low at 0.09. Is that reasoning correct? It would be helpful if the authors described in a short paragraph how they obtained one of the points in Fig. 1, panel A.

Agreed – now revised.

-p.2, line 30: “to successfully retrieve” → “to successfully retrieve it”.

Done

-p.2, line 36-37: “We found that retrieval satisfaction decreased with time since encoding and study session retrieval performance”. → “We found that retrieval satisfaction decreased with time since encoding and with lower study session retrieval performance”.

Done

-p.2, line 38: “But we also found … indicating” → “We also found … , indicating”.

Done

-p.5, line 105: “reward at play the presence of” → “reward at play in the presence”.

Done

-p.11, line 255: The data file in the OSF platform was uploaded with the .txt extension. It would be more helpful if it were uploaded with the the .csv extension, as the OSF system would be able to read the file and print it on the screen of the web browser.

We have now uploaded a .csv of the data to OSF to facilitate accessibility.

-p.12, line 289 (section “Encoding section results” and wherever applicable): The abbreviations for mean (M) and standard deviation (SD) should be reported in capital letters and italics. Please revise all such cases in the manuscript (e.g., p.15, section “Final recall test results”).

Thanks for the note. Revised throughout.

-p.13, line 309 (Figure 1 caption): The first letters should be capitalized in the boldface caption: “Study Session Learning and Ratings”. In addition, the legends in the x- and y-axis should have the first letter of each word capitalized: x = Retrieval Improvement; y = Learning-related confidence change. Same applies for panels B and C.

Revised

-p.13, line 317: References to Figures should be abbreviated (Figure 1 → Fig. 1; apply throughout the text for figure 1 → Fig. 1, figure 2 → Fig. 2, and figure 3 → Fig. 3).

Done

-p.15, line 362 (Figure 2 caption): The first letters are capitalized in the boldface caption: “Performance and Rating Associations”. The legends in the x- and y-axis should have the first letter of each word capitalized: x = Estimated performance (Rasch); y = z(Satisfaction Rating). Same applies for panels B and C.

Revised

-p.17, line 413 and line 415: Change the b subscripts to keep consistency with new terminology (performance memory → session memory), so that b_{session.perf} becomes b_{session.mem}. One case on p. 413, and the other on p. 415.

Done

-p.18, line 436: “z-scored JOL rating at study.” → “z-scored JOL rating unit at study.”

Done

-p.20, line 489: “…participants who did not complete the experiment were slightly more likely to be male …”. Shouldn’t it be female? On p. 11, line 262, the authors say that “Of those who did not complete all sessions, 20 identified as female and 11 as male”. So, the claim on p.20, line 489 should be “more likely to be female”.

Yes – thanks and revised

-p.20, line 499: “…, what then might the adaptive value … be?” → “…, what then might be the adaptive value …?”

done

-p.22, line 542: Here I am not sure which preposition is the most appropriate, as it depends on the meaning the authors wish to convey. The text reads “It is also possible that the learning reward argument in curiosity theory is incorrect or at least that the hedonic ‘wanting’ (Berridge & Kringlebcach, 2015) [Note: please use Plos One reference style here] aspect of reward is irrelevant in curiosity.” Shouldn’t it be “irrelevant to curiosity”?

revised

---

## [Editor Report · Decision Letter 2]

2 Oct 2023

Reliable retrieval is intrinsically rewarding:

Recency, item difficulty, study session memory, and subjective confidence predict satisfaction in word-pair recall

PONE-D-23-11730R2

Dear Dr. Holm,

We’re pleased to inform you that your manuscript has been judged scientifically suitable for publication and will be formally accepted for publication once it meets all outstanding technical requirements.

Kind regards,

Stergios Makris

Academic Editor

PLOS ONE
---

## [Editor Report · Acceptance letter]

10 Oct 2023

PONE-D-23-11730R2 

Reliable retrieval is intrinsically rewarding: Recency, item difficulty, study session memory, and subjective confidence predict satisfaction in word-pair recall 

Dear Dr. Holm:

I'm pleased to inform you that your manuscript has been deemed suitable for publication in PLOS ONE. Congratulations! Your manuscript is now with our production department. 

Kind regards, 

on behalf of

Dr. Stergios Makris 

Academic Editor

PLOS ONE